# Nitric Oxide as a Determinant of Human Longevity and Health Span

**DOI:** 10.3390/ijms241914533

**Published:** 2023-09-26

**Authors:** Burkhard Poeggeler, Sandeep Kumar Singh, Kumar Sambamurti, Miguel A. Pappolla

**Affiliations:** 1Department of Physiology, Johann-Friedrich-Blumenbach Institute for Zoology and Anthropology, Faculty of Biology and Psychology, Georg August University Göttingen, Zappenburg 2, D-38524 Sassenburg, Germany; 2Indian Scientific Education and Technology Foundation, Lucknow 226002, India; sandeeps.bhu@gmail.com; 3Department of Neurobiology, Medical University of South Carolina, 173 Ashley Avenue, BSB 403, Charleston, SC 29425, USA; sambak@musc.edu; 4Department of Neurology, University of Texas Medical Branch, 301 University Boulevard, Galveston, TX 77555, USA; pappolla@aol.com

**Keywords:** aging, amino acids, antioxidant, arginine, nitric oxide, oxidative stress, protein, peptide

## Abstract

The master molecular regulators and mechanisms determining longevity and health span include nitric oxide (NO) and superoxide anion radicals (SOR). L-arginine, the NO synthase (NOS) substrate, can restore a healthy ratio between the dangerous SOR and the protective NO radical to promote healthy aging. Antioxidant supplementation orchestrates protection against oxidative stress and damage—L-arginine and antioxidants such as vitamin C increase NO production and bioavailability. Uncoupling of NO generation with the appearance of SOR can be induced by asymmetric dimethylarginine (ADMA). L-arginine can displace ADMA from the site of NO formation if sufficient amounts of the amino acid are available. Antioxidants such as ascorbic acids can scavenge SOR and increase the bioavailability of NO. The topics of this review are the complex interactions of antioxidant agents with L-arginine, which determine NO bioactivity and protection against age-related degeneration.

## 1. Introduction

The anticipated demographic transition to an exponentially growing population of elderly persons with increased morbidity poses the greatest challenge to society in history. This challenge drives the following key scientific research question: Can we enhance human health span with the ever-increasing life expectancy resulting from advances in healthcare to prevent premature mortality? Aging is now the dominant risk factor for many degenerative disorders for which mechanisms and dietary or environmental modulators remain poorly studied. The progressive increase in health care costs for non-communicable conditions and the rise in morbidity and mortality with advanced age is promoted by the cumulative bioenergetic burden upon the target population by the Western diet rich in sugar, fat, and salt [1,2,3,4]. An imbalance between NO and SOR has been demonstrated in metabolic diseases such as diabetes and obesity [1,2,3]. Antioxidant protection can determine health [1,2]. Nutrition is decisive in determining health and healthcare costs [4]. Precision nutrition, specific food, and biomatrix supplementation have been proposed to improve health by supplying sufficient macro and micronutrients [5]. Adaptation and aging can be the opposite outcomes of dynamic developmental plasticity [6]. The discovery and development of effective supplements containing amino acids and antioxidants that can restore and improve health even at an advanced age is a rapidly developing field of applied biosciences [7]. Aging can be seen as a process of internal desynchronization induced by stress and aberrant-signaling-induced senescence and the concurrent loss of bioenergetic potential with a depletion of resources to prevent degenerative changes [8]. Supplementation can maintain or even improve human health [9]. This review focuses on the newest developments in supplementation with bioenergetic nutrients such as L-arginine and antioxidant agents such as B vitamins and vitamin C.

## 2. Adaptation and Aging

Lifetime exposure to high glucose and free fatty acid levels induces cumulative toxicity that limits adaptational and developmental plasticity [4]. Premature aging and disease can result from nutrition rich in calories but poor in nutrients and natural agents [9,10,11,12,13,14,15,16,17,18]. Supplementation rich in certain amino acids switches the metabolism to enhanced activity, efficacy, and oxidative phosphorylation capacity that improves mitochondrial redox regulation, inducing antioxidant adaptation by retrograde trophic pro-survival signaling [9]. Since caloric restriction is often associated with malnutrition in humans, only bioenergetic agents such as the mitogenic and mitotrophic amino acids glutamine, proline, and arginine, which are abundantly present in proteins and peptides from pulses, grains, or collagen, can significantly improve the metabolism of mitochondria and stimulate their signaling [9,10,13,14]. These amino acids are a real option to substantially extend the human health span [10,11,12]. L-arginine and L-arginine-rich proteins or peptides can supply the necessary nutrients to reduce glycemic load, insulin resistance, and lipotoxicity by facilitating and enhancing fat oxidation and reducing glucose accumulation [1,2,3,14]. Bioenergetic agents such as L-arginine and related amino acids have overall positive health effects [15,16,17], as demonstrated in the target population. These supplements induce bioenergetic stimulation, antioxidant protection, and ubiquitous regeneration that improve, restore, and maintain gut, skin, and joint health [10,11,12]. The synergistic effects of this unique L-arginine-rich blend with antioxidant agents of high bioenergetic potency are discussed in the context of easy-to-handle approaches in supplementation aimed at improving, regaining, or maintaining health by improving the diet of the target population.

## 3. Nutrition and Health

Food and supplementation can be a decisive factor in maintaining health during aging and stress or enhanced demand for protective nutrients [10,11,12,13,14,15,16,17,18]. Recent research indicates that a high intake of soy, pea, and pumpkin, rich in arginine, proline, and glutamine, can limit carbohydrate and fat toxicity associated with the Western diet and its predominant arginine-poor animal protein content [4,5,10,11]. The use of amino acids like L-arginine together with the synergistically acting B vitamins, folate, vitamin B_6_, and vitamin B_12_, in preparations of premium quality and natural origin, opens up new perspectives in establishing a molecular, metabolic medicine that enables prevention, therapy, and rehabilitation for improving, maintaining and, restoring the health of older adults [19,20,21,22,23,24]. Antioxidant protection and health-span extension seem to be possible through such an innovative approach as using amino acids and vitamins to enhance trophic retrograde NO signaling and thus life- and health-span [6,12,21]. This review reveals how a holistic strategy employing amino acids like arginine combined with other nutrients can reverse chronic degenerative changes and trigger adaptive reactions and repair processes that restore regeneration via redox regulation and antioxidant protection [6,12,21]. Novel, innovative approaches using highly sophisticated supplementation protocols have revealed the molecular mechanisms and physiological mediators of viability and survival that enable the organism to cope with internal and external stressors [6,12,21]. All molecular mediators that induce such adaptive plasticity act as mitochondrial metabolism modifiers to increase trophic support through the enhanced supply and more efficient use of bioenergetic resources [25]. The aim and goal of these approaches are to promote human fitness and health [22,23,24,25,26,27]. The universal bioenergetic decline as a hallmark of stress and senescence can be corrected through supplementation-dependent mitochondrial support that restores metabolic control mechanisms essential to regeneration and repair.

## 4. Say NO to Aging

Aging is often associated with increased adiposity and altered reduced muscle mass or sarcopenia, including increased ectopic fat stores such as visceral, hepatic, and intermuscular fat, which are independently associated with increased risk of cardiometabolic and physical dysfunction affecting gut mucosa, skin epidermis, and the entire myofascial system including the joints [1,2,3,14]. Precise supplementation for older adults should target the fat compartments while maintaining and even gaining muscle and lean body protein mass through a balanced approach providing maximal bioenergetic and cardiometabolic benefits by substantially improving metabolic control [1,2,3,22,23,24]. The development of insulin resistance is the decisive determinant for endothelial dysfunction associated with age-associated obesity and sarcopenia [1,2,3]. Recent studies have implicated the involvement of mitochondrial superoxide and peroxynitrite in inflammation [1,28,29,30,31]. As illustrated in Figure 1, asymmetric dimethylarginine (ADMA) and the L-arginine-depleting enzyme arginase mediate the insulin-resistance-induced reduction in NO formation and, thereby, the impairment of endothelium-dependent vasodilatation in human morbid obesity and sarcopenia [1,2,3,9,21,28,29].

The age-dependent enhancement of ADMA levels and the upregulation of L-arginine depletion through enhanced arginase activity are the decisive factors in the alteration of the L-arginine/NO pathway associated with insulin resistance and endothelial dysfunction [9,17,21,28,29,30]. These findings help to explain the profound effects of precision supplementation based on an enhanced supply of L-arginine and other bioenergetic agents in restoring metabolic control and reducing insulin resistance and lipotoxicity associated with enhanced superoxide anion radical and peroxynitrite formation as demonstrated in older adults [1,2,3,9,17,21]. Currently, many clinical studies are conducted to ensure that L-arginine supplementation and L-arginine-rich food can restore redox regulation in the elderly target population [9,21,28,29,30]. Aging decreases the arginine:ADMA ratio and the nitric oxide:superoxide ratio, leading to oxidative stress, inflammation, and degenerative changes that harm development and health [28,29,30]. Supplementation with amino acids such as L-arginine and L-arginine-rich food via certain peptides and proteins can restore a healthy arginine:ADMA ratio.

## 5. Aging and Oxidative Stress

As illustrated in Figure 2 and Figure 3, aging is associated with an increase in ADMA and a reduction in arginine that generate a dangerous shift in the ratio of arginine to ADMA and NO to SOR.

These ratios have to be shifted back and normalized through supplementation. The positive effects of biomatrix precision supplementation on gut, skin, and joint health can greatly improve older adults’ wellness and life quality [9,10,11,12,21,22,23,24,28]. These simple and feasible approaches can be easily and successfully implemented and are safe, since they are based on traditional food and food ingredients with a well-established safety profile and excellent tolerability, as illustrated and demonstrated in Figure 4.

The homeostatic model assessment of insulin resistance (HOMA IR) index reveals that the development of insulin resistance leads to glucose intolerance despite quite normal fasting glucose levels and HbA1c levels within the normal range in the elderly target population [1,2,3,7,14,17]. The elderly population in Germany was considered to be healthy, although they exhibited a strong decline in arginine:ADMA and nitric oxide:superoxide ratios [9,21,28].

The age-dependent increases in homocysteine and ADMA lead to enhanced endogenous “inflammaging”. The low NO/O_2_- ratio can only be reverted by targeting insulin dysfunction, cumulative lipotoxicity, and endothelial dysfunction [1,2,3,14,17,29,30,31]. The conclusive data and findings on arginine-based antioxidant supplementation indicate that such approaches aimed at regaining metabolic control and restoring health in the elderly are urgently needed and well tolerated [9,21,28,29,30].

This review collects and extends these findings, demonstrating the necessity of using arginine-based protective supplementation to improve our nutrition by providing sufficient NO levels [1,2,3,14,17,25,26,27,28,29,30,31,32,33,34,35]. L-arginine and L-arginine-rich food can maintain blood flow, cardiovascular health, cognitive performance, metabolic regulation, body muscle mass, and body composition [7,21,22,23,24].

NO is an endogenous metabolic mitochondrial master modulator that mediates antioxidant protection and regeneration [9,21]. The balance of pro- and antioxidant factors is shifted towards the prooxidant radicals during aging but can be maintained through supplementation, in older people, with L-arginine or L-arginine-rich food [10,11,12,21,22,29,34,35,36]. The upregulation of the iron–sulfur complex N2 in complex I and cytochrome-c-oxidase subunit 5A in complex IV may be the decisive mechanism of NO-mediated antioxidant protection [21,26,27]. NO mimetics and NO donors also target these metabolic master mechanisms [27]. The supply of antioxidants that scavenge the superoxide anion radicals is a reasonable strategy to amplify the effects of L-arginine and L-arginine-rich food. These compounds also increase the bioaccessibility and bioavailability of L-arginine as the substrate of NO formation [21].

## 6. Chrononutrition and Synchronisation

NO determines mitochondrial biogenesis and bioactivity [32]. The master regulator of energy metabolism is also the key factor of neurovascular-neuro energetic coupling and a core element of synaptic plasticity and, thus, brain development and cognition [33]. The superoxide anion radicals are antagonistic mediators that can induce neurodegeneration and cell death by increasing oxidative stress and damage directly and by depleting NO and L-tryptophan associated with a lack of the L-tryptophan-related antioxidants melatonin, indole-3-propionic acid and indole-3-propionamide [25,26,27,28,29,33,34,35,36,37]. Furthermore, melatonin and such indoles can decrease the amyloid burden and toxicity by facilitating the removal of the reactive abeta peptide from the brain through glymphatic drainage [38,39].

L-tryptophan, serotonin, N-acetylserotonin, and melatonin are powerful antioxidant and bioenergetic agents and act as neurotrophic compounds that protect the brain against toxic insults such as oxidative stress [25]. Likewise, indole-3-pyruvic acid, indole-3-propionic acid, and indole-3-propionamide act as protective L-tryptophan metabolites in the gut and brain [25,36,37]. The L-tryptophan oxidation product of the reaction of superoxide anion radicals with L-tryptophan, L-kynurenine, is an agonist at the AHR (aryl-hydrocarbon) receptor and, as such, a strong prooxidant and potent neurotoxic agent [35,36,37]. L-kynurenine formation promotes superoxide anion radical generation, thereby increasing non-enzymatic and enzymatic tryptophan degradation. Thus, under such prooxidant conditions and stress, the activity of the gastrointestinal enzyme, indoleamine-2,3-dioxygenase (IDO), is significantly enhanced, and a vicious cycle of proinflammatory and potentially neurotoxic conditions and reactions is initiated and sustained [35,36,37].

L-arginine and NO can prevent this dangerous situation via antioxidant protection and superoxide detoxification, not only in the gut but also in the entire organism, including the brain [25,26,34,35,36,37,38,39]. The amino acids, L-arginine and L-tryptophan, can act together to prevent oxidative stress and damage, inducing potent neuroprotection [17,21,25,26,35,37]. Preliminary findings indicate that protective L-arginine supplementation can enhance human circulating melatonin, indole-3-propionic acid, and indole-3-propionamide levels [7,17,25,26,33,37,38,39]. L-arginine and L-tryptophan can be provided in sufficient amounts by food, food supplements, food for specific medical purposes, and nutraceuticals [7,17,25,26]. The glymphatic system can reduce the amyloid burden by providing protective indole agents that are increased by L-arginine and L-arginine-rich food [7,17,25,26,27,28,29,30,31,32,33,34,35]. Melatonin has been demonstrated to facilitate amyloid removal via drainage through this detoxification system [38,39].

The neurovascular–neuroendocrine–neuro energetic coupling is enhanced by amino acids such as L-arginine and L-tryptophan, as well as by many of their endogenous metabolites [25,26,33,37,38,39]. L-arginine and NO can enable detoxification, regulation, and regeneration [7,17,21,33,38,39]. The digestive tract connects arginine and tryptophan metabolism and is central in mediating the potent anti-aging effects of augmented NO signaling [34,35,40,41,42,43]. L-arginine and L-arginine-rich food may act as enteronutrition, immunonutrition, and chrononutrition that target the second brain, the gastrointestinal system, and the third brain, the resident symbiotic organisms therein [40,41,42,43]. L-arginine and L-arginine-rich food can prevent the consumption of L-tryptophan with its irreversible oxidation to L-kynurenine [34,35,43]. The pleiotropic response to L-arginine orchestrates a broad range of protective reactions and pathways that converge on L-tryptophan and its derivatives, as illustrated in Figure 5 and Figure 6.

L-arginine is needed to restore a healthy ratio of L-tryptophan: L-kynurenine in older adults [21,40,41,42,43]. Food rich in L-arginine and antioxidants can exert potent anti-aging effects that increase health and fitness. Biomatrix precision supplementation uses nutrients that target the decisive metabolic pathways [40,41,42,43,44,45,46,47,48]. These are the arginine and tryptophan pathways, and they have to be rerouted [49,50,51,52,53,54,55,56,57,58,59,60]. The gut–brain connection, with protective signaling by symbiotic organisms in the digestive tract, seems to be decisive in this regard [43]. The potent effects on health are primarily mediated by protective antioxidant indoles such as melatonin, indole-3-propionic acid, and indole-3-propionamide that originate in large parts from symbiotic organisms in the gastrointestinal tract [43]. As mitochondrial metabolism modifiers, they prevent the formation of harmful superoxide anion radicals that destroy nitric oxide, uncouple its biogenesis, and reduce its bioavailability and bioactivity [43]. The indoles act as potent radical scavengers and endogenous protective agents. Catalytical antioxidants can avoid superoxide anion radical formation during oxidative phosphorylation in mitochondria and act as preventive catalytic bioenergetic agents. This lowers the burden of oxidative stress and damage largely due to mitochondrial superoxide anion radical formation [1,21,26]. The protective NO radicals can thus benefit human health in concert with the indole bioenergetic agents that improve metabolic control and mitochondrial energy metabolism efficacy [1,21,22,25,26,43].

## 7. Determinants of Life and Health Span

Recent research has confirmed the decisive roles of these metabolic pathways in regulating and determining human health [40,41,42]. Exploring new avenues that enable active living and healthy aging by preserving our fitness throughout our entire lifetime is the ultimate aim and goal of ongoing research in the field [43,44,45,46,47,48]. The upregulation of NO bioavailability can prevent premature aging and neurodegeneration characterized by cognitive decline [49,50,51,52,53,54,55,56,57]. Recent research has confirmed that arginine-rich protein from lupin, pea, or buckwheat can enhance circulating levels of indole-3-propionic acid in humans [58,59]. Tryptophan-derived antioxidant indole acids, amides, and esters can exert potent health-promoting and neuroprotective effects similar to regenerative L-arginine and NO signaling [43,60].

The non-enzymatic NO formation via nitrate and nitrite also contributes to a sufficient supply and the health promoting effects of the gasotransmitter [50,51]. Boosting this endogenous nitrate-nitrite-NO pathway has been conclusively demonstrated to improve gastrointestinal, cardiovascular, metabolic and cognitive performance both in humans and in animal models of disease [50]. Nitrate-derived NO has been shown improve several physiological functions that typically decline during aging and thus, the simple supplementation of the diet with nitrate can improve health, fitness, and performance in the elderly [50]. NO adducts formed after enhanced formation of NO induced by nitrate can lead to sustained improvements of metabolic disorders [50]. Vegetables rich in nitrate, such as spinach and beetroot, are a good source of NO, with beneficial effects on validated markers of cardiovascular health and an association with a lower risk of cardiovascular disease [51]. Given the association between cardiovascular disease risk factors and dementia and the important role of NO in vascular health and cognition, it is likely that dietary nitrate could also improve cognitive function, markers of brain health, and lower risk of dementia. The positive effects of nitrate supplementation on cardiometabolic regulation, neurovascular coupling, and cognitive performance indicate that nitric oxide plays a decisive role in enabling healthy aging.

L-arginine’s potent health-promoting effects are conclusively demonstrated [9,22,23], although many open questions have to be addressed. The beneficial cardiovascular effects of L-arginine have been established and confirmed [9,15,16,22,23,47], but the potent cognition enhancing effects have only been reported in a small clinical pilot study [60]. L-arginine and B vitamins have been demonstrated to be of great importance in maintaining and improving health [20,21,22,23,24,61,62,63,64,65,66]. The enhanced levels of prooxidant kynurenines and indoles increase superoxide anion radical generation and impair NO formation [67]. Adipocyte-derived kynurenine induces insulin resistance and a metabolic syndrome that can be antagonized by B vitamins such as B_6_ [68].

The age-dependent decline of tryptophan in the brain is associated with the enhanced formation of toxic kynurenines [69]. This demonstrates a bidirectional interaction of the arginine and tryptophan metabolic pathways that determine NO or superoxide anion radical levels [67,68,69]. Without sufficient nitric oxide, L-tryptophan is metabolized largely to *N*-formyl-l-kynurenine and L-kynurenine by indoleamine-2,3-dioxygenase 1 and 2 and by tryptophan-2,3-dioxygenase. Their age-dependent increase leads to arginine, tryptophan, and antioxidant indole depletion, as well as toxic indole and kynurenine accumulation, exerting potent prooxidant and proinflammatory effects [67,68,69].

As elevated levels of L-kynurenine are associated with insulin resistance and the metabolic syndrome, arginase activity is induced, and ADMA degradation by the enzymes, dimethylarginine-dimethylaminohydrolase (DDAH), with the two DDAH isoforms, DDAH 1 and DDAH 2, is reduced [17]. Thus, the substrate of NO formation, L-arginine, is depleted, and elevated ADMA levels induce the uncoupling of NO synthase with the enhanced appearance of superoxide anion radicals due to oxidative stress as a consequence [1,9,17,21,22,23,28,30,33]. Since the cofactor tetrahydrobiopterin is oxidized and depleted, a dangerous shift is induced from antioxidant protection to oxidative stress and damage [21,30,34,35,61,62].

The balance of NO to superoxide anion radicals is shifted to the prooxidant phenotype seen in premature aging or cardiometabolic, cardiovascular, and cerebrovascular diseases [1,17,34]. Supplementation with L-arginine and the B vitamins B_6_, folic acid, and B_12_ can stop the vicious cycle [22,23]. Thus, tryptophan levels have to be selectively increased, for instance, by L-arginine or L-arginine-rich food that contains sufficient B vitamins, leading to enhanced formation of the antioxidant indole agents to boost NO bioactivity and bioavailability [22,23,43,44,45,46,47,58,59]. Supplementation and dietary management of age-associated diseases target elevated blood pressure in the normal and high-normal range due to reduced endothelium-derived NO formation, reduced blood flow by improving endothelium-dependent NO-mediated vasodilation in the early stages of arteriosclerosis, and enhanced homocysteine levels by supplying sufficient folate, vitamin B_6_, and B_12_ as needed [22,23,43,44,45,46,47,58,59]. Such supplements and medical food can prevent premature aging and enable healthy aging.

## 8. Nitric Oxide Boosts Protective Tryptophan Pathways

The studies on L-arginine-rich food have severe limitations due to size and scope, and they all lack the necessary power to provide proof of principle [44,47,58,59]. The small size of these studies does not allow us to arrive at any conclusion on the clinical relevance of augmented NO signaling, nor are the decisive molecular mechanisms and mediators of such supplementation fully identified and sufficiently characterized, pointing to the urgent need to conduct broad, large-scale clinical trials that address the complexity of the metabolic pathways involved in regulating NO bioavailability and bioactivity. The decisive contribution of these metabolic pathways in determining the ratio of NO to superoxide anion radicals and, thus, the balance of regeneration to degeneration has, however, been firmly established [41,64,67].

L-arginine and L-arginine-rich food can reroute tryptophan metabolism from prooxidant indoles and kynurenines to antioxidant indole agents that restore a healthy nitric oxide: superoxide anion radical ratio [21,41,43,48]. A relative disease or age-dependent L-arginine deficiency and L-tryptophan depletion can cause significant oxidative stress and degeneration [34,41,65]. The enhanced consumption of the substrate of NO formation, L-arginine, and the cofactor, tetrahydrobiopterin, induces a higher demand for B vitamins such as B_6_, folic acid, and B_12_ [21,22,36,37,67,68,69].

A biomatrix precision supplementation can provide all these nutrients that target the lower gastrointestinal tract with its resident symbiotic organisms to supply sufficient amounts of amino acids, antioxidants, and B vitamins [59]. This will restore an adequate supply of nitric oxide and increase tryptophan levels and synthesis of the protective antioxidant indole agents, melatonin, indole-3-propionic acid, and indole-3-propionamide [43,45,59]. Current ongoing studies focus on approaches that utilize the potential of protective antioxidant scavenging and signaling by NO to significantly extend human life and health span [64].

## 9. L-Arginine and B Vitamins Extend Health and Life Span

NO and superoxide anion radicals are antagonistic key players in shaping aging and adaptation [1,21,64]. Potent beneficial effects of L-arginine and B vitamins have been demonstrated to maintain or even improve human cardiometabolic and cardiovascular health [22,23,61,62,63,64,65,66]. They exert antioxidant, anti-inflammatory, and antiaggregatory effects that can limit or even reverse premature aging and degenerative diseases [22,23,41,45,46,59,60,61,62,63,64,65,66]. The supplementation of L-arginine and L-arginine-rich food with the B vitamins, vitamin B_6_, folic acid, and vitamin B_12_, allows for the successful dietary management of the age-dependent cardiometabolic diseases by assuring a sufficient supply of NO [20,21,22,23,24,61,62,63,64,65,66]. Such foods for specific medical purposes (FSMP) are authorized for the dietary management of NO deficiency associated with enhanced blood pressure, reduced blood flow, and hyperhomocysteinemia [22,23]. The age-dependent tryptophan degradation to kynurenine and the accumulation of toxic kynurenine metabolites, such as quinolinic acid, as largely mediated by prooxidant and proinflammatory superoxide anion radicals, can be prevented [34,35,36,37,40,41,42,64]. SORs consume the cofactor of NO synthesis, tetrahydrobiopterin, and oxidizes tetrahydrobiopterin to biopterin and neopterin, which are both inactive [34,35,36,37,40,41,42,64,69]. Folic acid is needed to recycle tetrahydrobiopterin, and this B vitamin and its derivatives can also act as cofactors for NO synthesis and synergistic induction of the formation of this protective radical [22]. The insufficient supply of B vitamins impairs NO synthesis, and the generation of this protective agent is uncoupled and disrupted [22]. Instead of nitric oxide, superoxide anion radicals are generated and induce oxidative stress, damage, and NO depletion.

The SORs consume NO that scavenges these reactive intermediates, and the peroxynitrite formed in the process can decompose to highly reactive hydroxyl, carbon dioxide, and nitrogen dioxide radicals [21,64]. Since melatonin, indole-3-propionic acid, and indole-3-propionamide can provide potent neuroprotection by acting as catalytical antioxidants and powerful radical scavengers, the preventive and therapeutic potential of L-arginine and L-arginine-rich food and the antioxidant B vitamins should be further explored in future clinical trials [21,43,46,47,58,59]. Antioxidant protection can prevent disease- and age-related oxidative stress and damage [21,25,26,27,43]. NO seems to have a key role, and NO’s interaction with SOR is decisive [64]. Thus, it seems prudent to boost antioxidant and protective NO formation and reduce superoxide-anion-radical-mediated oxidative stress and damage by the key nutrients, L-arginine and the B vitamins [22,23].

NO contributes to vascular health and can reverse endothelial dysfunction and arteriosclerosis [62,64]. This may limit age-dependent degenerative processes and restore the supply of oxygen and nutrients to repair and regenerate the body and brain [22,55,64]. NO-boosting food and nutraceuticals improve cardiometabolic, cognitive, and neurovascular health [21,22,23,24,32,33,34,35,45,46,47,58,59,60,61,62,63,64,65,66]. Thus, as Hippocrates stated, food shall be your medicine and medicine your food. These approaches may add years to life and life to years. Their implementation is of the utmost importance and relevance to translational medicine and the life sciences that enable progress in healthcare. Future studies will also target new molecular mechanisms and mediators that extend life and health span and allow for exceptional longevity associated with good health.

## 10. The Need for Sufficient NO Formation

Because of its pleiotropic nature, NO can become cytotoxic at high concentrations [70]. The antioxidant effects usually prevail since they are broad and include direct radical scavenging and antioxidant protection against reactive intermediates that induce oxidative stress and damage [71,72]. Excessive concentrations of NO can be generated with high concentrations in the central nervous system and exert detrimental effects, particularly in depression [73]. An unfortunate interaction with antidepressant drugs should be avoided [73]. At physiological concentrations of L-arginine, such as those provided with foods, food supplements, and foods for specific medical purposes, no excessive formation of NO is expected since the second-generation L-arginine supplements contain less than 3 g of L-arginine [21,22,23,45,46,47].

Nevertheless, the administration of foods for specific medical purposes requires consultation with a physician that approves their use in the EU. Subjects with kidney problems should exert caution when supplementing proteins and are advised to adhere to a low- or ultra-low-protein diet. However, the moderate consumption of vegan L-arginine-rich proteins seems beneficial rather than detrimental to this specifically vulnerable group of patients, and plant-based diets have been demonstrated to be superior compared to animal-based diets [74]. Plant proteins are less likely to induce glomerular hyperfiltration than animal proteins. Protein supplementation is recommended when these patients are affected by sarcopenia [74]. Non-animal protein does not lead to hyperphosphatemia, as previously believed. Evidence indicates that plant protein is protective against kidney disease [74]. Supplementation should always address the specific needs of the target population. Older adults generally demand L-arginine and L-arginine-rich proteins to improve and maintain adequate NO formation [9,59,64]. Patients with cardiovascular and cardiometabolic diseases also have an enhanced need for L-arginine and L-arginine-rich proteins to restore and sustain a healthy NO supply [21,22,23,45,46,47]. Although vegan proteins contain higher amounts of L-arginine than animal proteins, animal proteins can restore sufficient L-arginine levels to improve and maintain NO formation and blood vessel dilatation with enhanced blood flow [75,76,77]. L-arginine and related amino acids can restore, improve, and maintain health [78,79]. L-citrulline is an attractive alternative to L-arginine and be used together or instead of L-arginine [80]. L-homoarginine has also recently been demonstrated to exert potent protective effects that maintain high NO levels for healthy aging [81].

Recently, conclusive evidence for the decisive role of L-arginine in preserving brain health has been summarized and has demonstrated that NO could prevent cognitive impairment [82]. Amyloid accumulation, for instance, causes disturbances in the arginine and tryptophan metabolism that can be prevented by antioxidants such as vitamin C [83]. Arginase 1 and indoleamine 2,3-dioxygenase 1 (IDO1) are immunoregulatory enzymes which catalyze the degradation of L-arginine and L-tryptophan, respectively, resulting in amino acid deprivation [84]. Aging and age-related cardiovascular diseases lead to arginine and tryptophan depletion, impairing neurovascular coupling [85]. Cardiovascular disease is responsible for more deaths worldwide than any other type of disorder, and atherosclerosis with endothelial dysfunction is the cause of several kinds of cardiovascular disease [85]. L-arginine and L-tryptophan determine disease development and progression. The regulation of amino acid metabolism by indoleamine 2,3-dioxygenase (IDO) and arginases 1 and 2 is mediated through various signals and enzymes that can be targeted to treat and prevent atherosclerosis and cerebrovascular diseases, as outlined in this review [85]. L-arginine-rich collagen peptides have been shown to delay skin, joint, and muscle aging [86,87,88].

The emerging approaches to the prevention and treatment of aging and age-related diseases based on enhanced NO formation are promising. Healthy aging is possible and means a high life quality even at an advanced age. Proteins from non-animal sources cannot only decrease mortality but also morbidity and, thus, prevent premature aging and associated diseases [89,90]. As the substrate for NO formation, L-arginine is the decisive molecule that mediates non-animal proteins’ health-promoting effects. Vegan proteins also deliver an ample supply of glutamine- and proline-rich peptides. This safeguards a sustained and sufficient supply of L-arginine. Thus L-arginine supplied with amino acids, peptides, or proteins can assure a healthy nitric oxide: superoxide anion radical ratio to increase life and health span.

## 11. Conclusions

NO is the key compound of antioxidant protection that allows for neurovascular coupling, bioenergetic stimulation, and ubiquitous regeneration, enabling longevity in good health. Supplementation with L-arginine and L-arginine-rich- food assures sufficient NO synthesis. The age-dependent accumulation of its antagonist, ADMA, with the enhanced formation of dangerous superoxide anion radicals and decoupling can be neutralized by the administration of L-arginine or L-arginine-rich food with supplementation of proteins and peptides that contain large amounts of this amino acid. Future research could consist of investigating the best-targeted approaches to supply these nutrients to improve cognition and health. Redox regulation can restore a healthy ratio between the dangerous superoxide anion radicals and the protective nitric oxide. This antioxidant protection orchestrates bioenergetic signaling that assures adaption and synaptic plasticity and removes misfolded proteins that generate oxidative stress and damage in the brain. Future controlled interventional studies on antioxidant nutrients and their impact on health and life span are urgently needed to characterize the molecular mechanisms and mediators determining adaption and development.

## Figures and Tables

**Figure 1 ijms-24-14533-f001:**
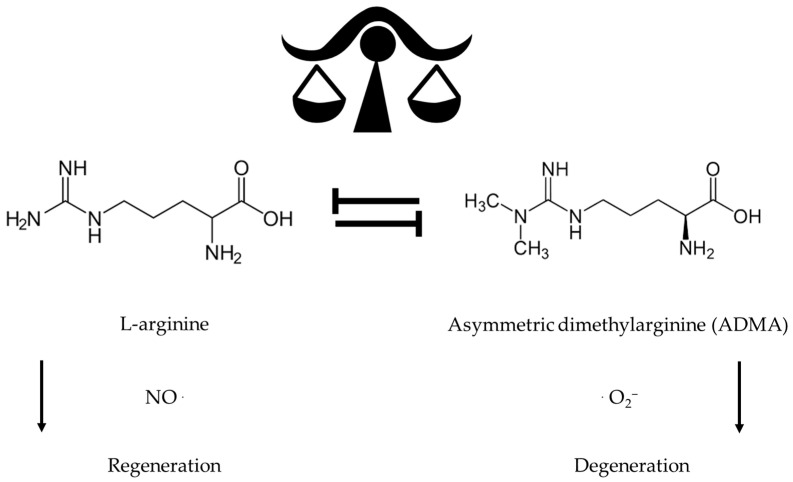
NO and superoxide anion radicals are decisive factors for regeneration and degeneration.

**Figure 2 ijms-24-14533-f002:**
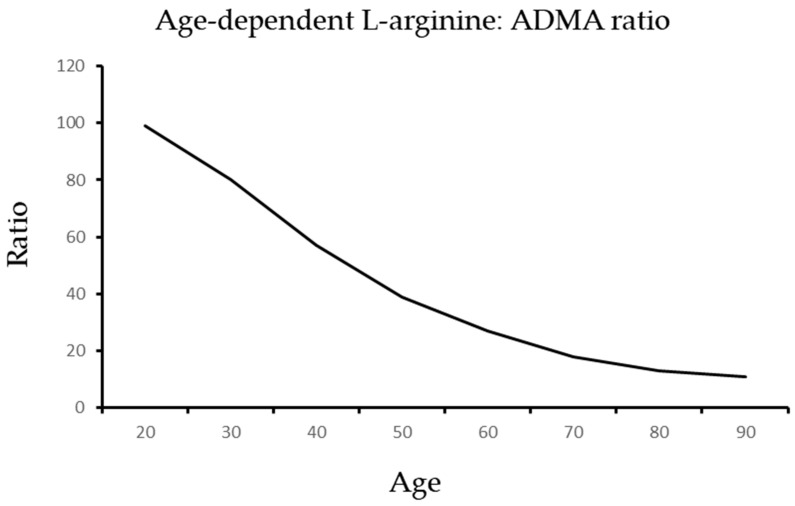
The age-dependent decline of the L-arginine: ADMA ratio.

**Figure 3 ijms-24-14533-f003:**
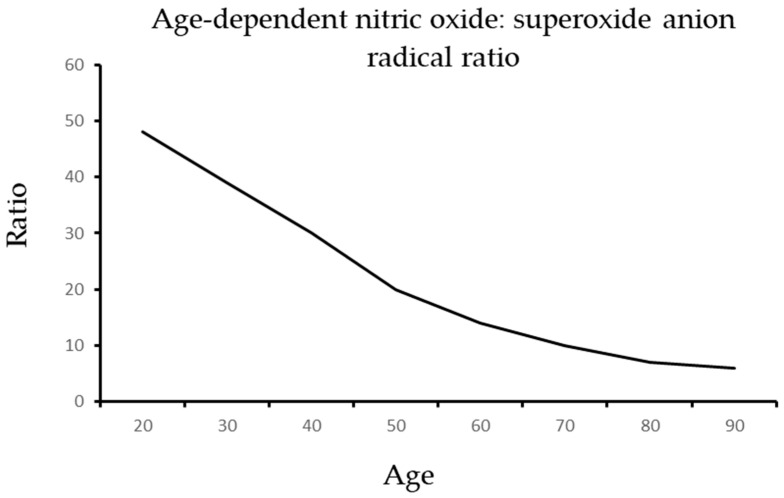
The age-dependent decline of the nitric oxide: superoxide ratio.

**Figure 4 ijms-24-14533-f004:**
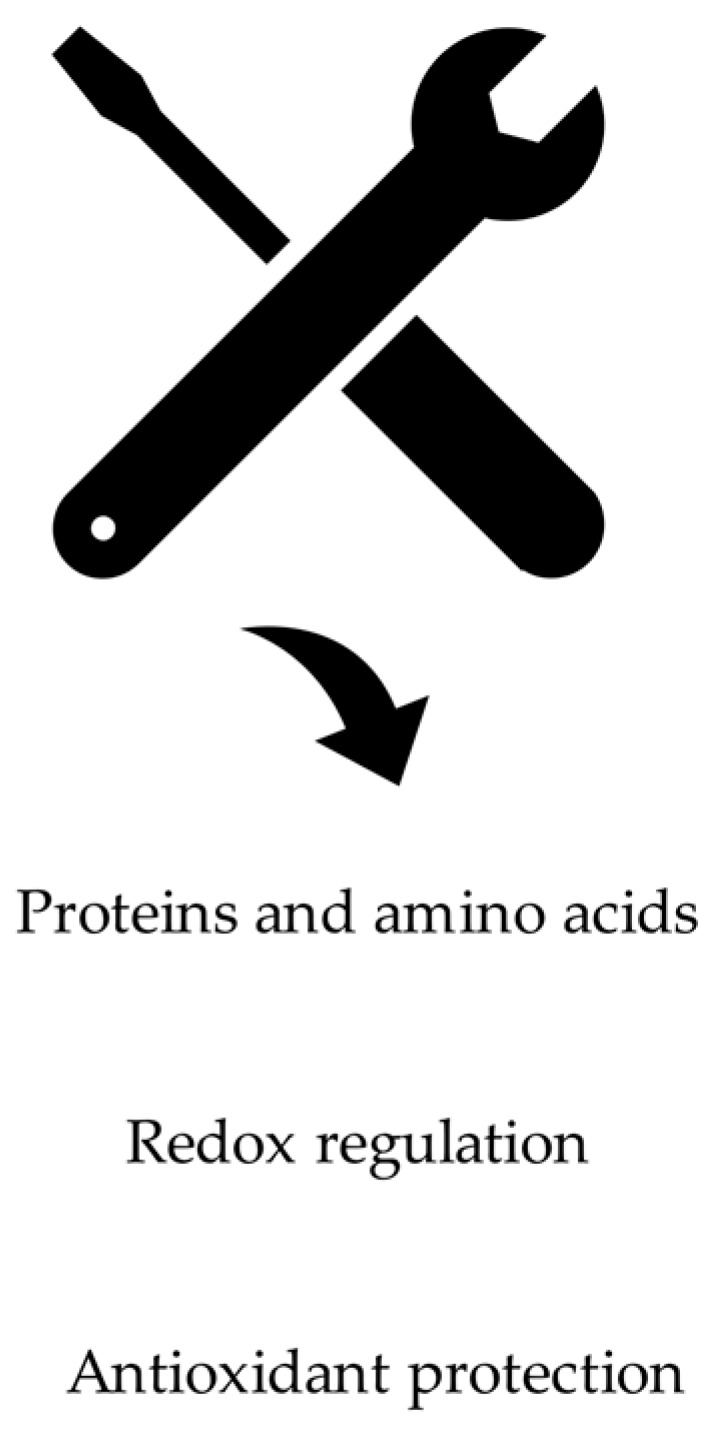
Amino acids such as L-arginine enable redox regulation and antioxidant protection.

**Figure 5 ijms-24-14533-f005:**
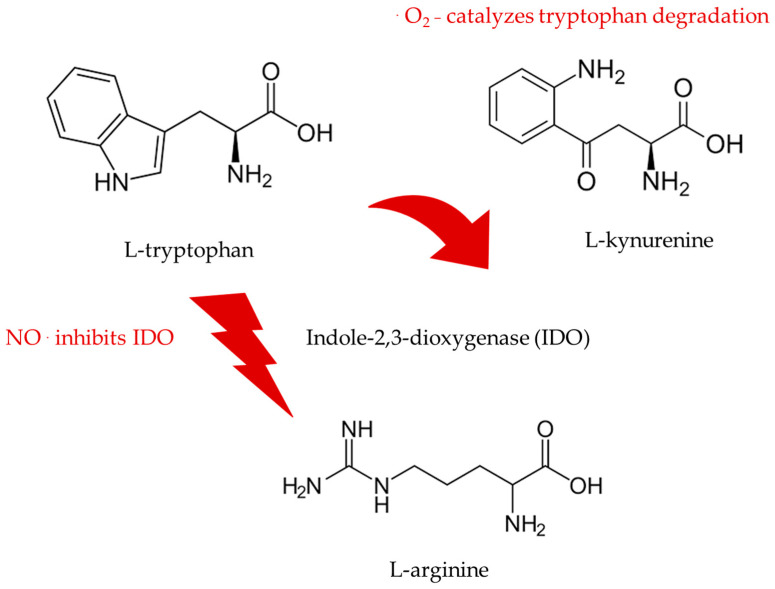
L-Arginine inhibits the enzyme indoleamine-2,3-dioxygenase via NO formation.

**Figure 6 ijms-24-14533-f006:**
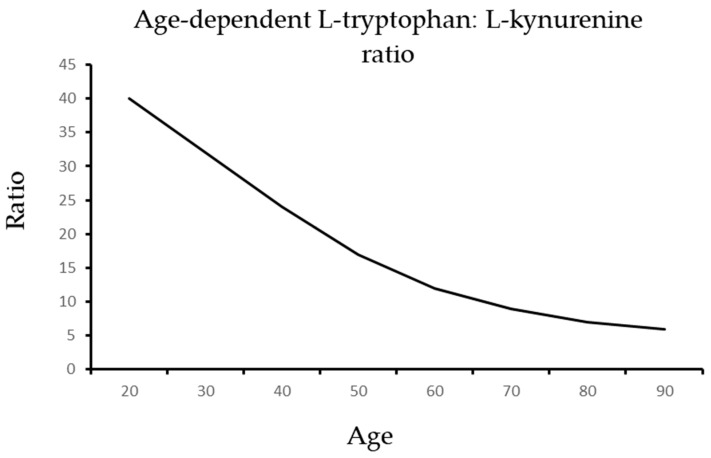
The age-dependent decline of the L-tryptophan: L-kynurenine ratio.

## Data Availability

Not applicable.

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
