# Peer review of "Nitric Oxide as a Determinant of Human Longevity and Health Span"

_ijms, 2023, doi:10.3390/ijms241914533_

Round 1

Reviewer 1 Report

The author reviewed that longevity and health are influenced by the interaction between nitric oxide (NO) and superoxide anion radicals (SOR), and that the amino acid L-arginine, along with antioxidant supplementation, plays a crucial role in balancing these factors to promote healthy aging and protect against age-related deterioration. The text explores how L-arginine and antioxidants, such as vitamin C, can increase NO production and bioavailability, and how this can contribute to maintaining health as we age. It also mentions the importance of displacing asymmetric dimethylarginine (ADMA) from the site of NO formation with L-arginine and how antioxidants can assist in this process. In summary, the balance between NO and SOR, along with appropriate supplementation, is essential for health and longevity.

The manuscript is well written and is of scientific importance.

In my opinion, the style of the figures in it does not honor the good work of the manuscript.

The figures that present pathways, such as Figures 1, 4, and 5, should be improved using some software (I recommend biorender) to give them neatness and clarity.

The figures that present a graph (figures 2, 3, and 6) should include the Y axes. Although they are understood, adding two Y axes (left and right) would help understand the figure.

In addition to the above, each figure should have a small legend that explains it in detail. I know the explanation is exposed in the text, but it would be helpful to add a description to improve its understanding.

Author Response

Thank you very much for your positive response and for your comments on the figures. The figures had to be improved and we used software to give them neatness and clarity. Symbols and pictures have been added to the figures that present pathways. Furthermore, when necessary we added small legends to explain what is depicted like in figure 5. Figure 1 and 4 should be self explaining now that they have the pictograms that are easily understood. The figures on the ratios should really reflect the ratios since they are important to fully understand the age-dependent changes. The figures now includes the Y axes and this is the ratio. There was also a title added to each of these figures that explains what is depicted. Each figure has also an x axes and a legend, here it is the age. This hopefully will resolve the critical issues of concern that you have addressed. Our take home message is like you said that the relative changes as demonstrated in the ratios may be decisive. This is now clearly depicted and illustrated by the figures. The balance in figure 1 symbolizes this. Thank you for giving us the opportunity to make these changes and additions to our manuscript in its revision.

Reviewer 2 Report

Although the authors write a review mainly focusing on enzymatic NO production via L-arginine supplementation affecting human longevity and health span, in particular when considering nutritional aspect and ageing with increasing vascular endothelial dysfunction, I think that non-enzymatic NO production via the dietary nitrate/nitrite/NO pathway should also be included and mentioned in detail in this review. Otherwise, I think that, as well as the text, the title should be changed with something clearly referring to and limiting to endogenous NO production from L-arginine supplementation.

The figures are too simple to comprehensively represent the contents of the text. I think they could be done a little better by adding images instead of just text, for example.

Author Response

Thank you very much for pointing out the importance of non-enzymatic NO formation via the dietary nitrate/nitrite/NO pathway. This important pathway is now considered. A whole new paragraph was added to demonstrate the relevance of this pathway. The nitrate/nitrate/NO pathway demonstrates that it is nitric oxide that mediates the positive effects of L-arginine on health and aging. Yes, the figures are really to simple and we have revised them incorporating images, symbols and short legends to better explain what is depicted. Futhermore, we added easy to understand pictograms for better understanding. Since we stress that fact that the relative age-dependent changes as seen in the ratios are decisive, we now clearly present our take home message in the figures.

Round 2

Reviewer 2 Report

The author answered my questions accurately. I have no objections to the revised version.